# IMPROVING FAIRNESS VIA NOISE INJECTION IN VISION TRANSFORMERS

**Qiaoyue Tang**[1]*, **Sepidehsadat Hosseini**[2], **Mengyao Zhai**[2], **Thibaut Durand**[2], **Greg Mori**[2,3]
[1]University of British Columbia, Vancouver, Canada
[2]RBC Borealis, Vancouver, Canada
[3]Simon Fraser University, Vancouver, Canada
qiaoyuet@cs.ubc.ca
{sepid.hosseini,mengyao.zhai,thibaut.durand,greg.mori}@borealis.com

## ABSTRACT

This paper presents **FairNVT**, a lightweight debiasing framework for pretrained transformer-based encoders that improves both representation and prediction level fairness while preserving task accuracy. Unlike many existing debiasing approaches that address these notions alone, we argue that they are inherently connected: representations that strongly encode sensitive information make prediction-level fairness fragile, while suppressing sensitive information at the representation level can facilitate fairer and more robust predictions. Our approach learns task-relevant and sensitive subspaces via lightweight adapters, applies calibrated Gaussian noise to the sensitive subspace in a randomized-smoothing style, and fuses it with the task representation; together with orthogonality constraints and demographic-parity regularization, these components jointly reduce sensitive-attribute leakage in the learned embeddings and encourage fairer downstream predictions. The framework is compatible with a wide range of pretrained transformer encoders. Across three datasets spanning vision and language, FairNVT reduces sensitive-attribute attacker accuracy, improves demographic-parity and equal-opportunity metrics, and maintains high task performance.

## 1 INTRODUCTION

Foundation models trained on large-scale, uncurated data have transformed visual and multi-modal learning through their transferable representations. Yet these models often encode *social and demographic biases* present in their training data, leading to systematic unfairness across attributes such as gender, race, and age. When deployed in sensitive domains, such as recruitment, credit scoring, or facial recognition, these biases result in inequitable treatment of individuals and undermine reliability of deployment.

Fairness in machine learning is commonly studied at the *prediction-level*, where the goal is to ensure that model outputs satisfy group-based criteria such as demographic parity, equal opportunity, or equalized odds. While prediction-level metrics are essential for measuring downstream impact, they provide only a partial view of the problem. While effective for a specific target task, this narrow focus often leaves the model's internal representations "biased", where latent embeddings frequently retain high mutual information with sensitive attributes, even if the final classifier appears fair. This creates a significant vulnerability: sensitive attributes may be easily recoverable from embeddings, enabling downstream misuse, model inversion, or fairness degradation (Feng et al., 2023; Gallegos et al., 2024). These observations motivates *representation-level* fairness, which aims to learn representations that is invariant, or at least weakly informative, with respect to the sensitive attributes to mitigate downstream discrimination.

In this paper, we argue that *representation-level fairness is not merely a secondary objective, but a foundational requirement that potentially improves and stabilizes prediction-level fairness*. By explicitly disentangling and "noising" the representation space, we obfuscate the underlying signals

---

*Work done during internship at RBC Borealis.

that the classifier uses to make biased decisions. When the latent space is demonstrably purged of sensitive information, the downstream prediction head is forced to rely on task-relevant features, leading to a more robust and naturally fair decision-making process.

Despite such connection, most existing methods focus on one level of fairness in isolation. Methods targeting prediction-level fairness typically impose constraints or regularization directly on classifier outputs (Kang et al., 2022; Wang et al., 2023; Xie et al., 2024), without explicitly addressing sensitive information encoded in representations. Conversely, representation-level methods often rely on adversarial (Zhang et al., 2018; Götte, 2023) or contrastive learning objectives (Park et al., 2022), or deploy projection-based removal (Islam et al., 2024; Shi et al., 2024) to reduce sensitive information leakage, but rarely examine how such interventions translate into improved prediction-level fairness. Moreover, these techniques often require unstable adversarial min–max optimization, repeated projection steps, or full fine-tuning of large backbone models, limiting their scalability and practicality for modern pretrained transformers.

To address these limitations, we revisit representation-level fairness from a *robust learning perspective* and propose **FairNVT**, a lightweight debiasing framework for pretrained transformer-based encoders built on randomized smoothing. By injecting calibrated Gaussian noise into the sensitive embedding subspace, the proposed method obfuscates sensitive information while preserving task-relevant cues, effectively connecting robustness and fairness. Unlike prior approaches that focus solely on prediction-level objectives, our framework explicitly mitigates sensitive-attribute leakage in the learned embeddings, resulting in improved fairness both in representation and downstream predictions. Empirically, we demonstrate that FairNVT achieves a favorable fairness–utility balance: sensitive-attribute predictability from debiased embeddings is substantially reduced, while task accuracy remains high and prediction-level fairness is consistently improved. This dual-level fairness makes the learned representations more reliable for downstream use.

## 2 PRELIMINARIES: FAIRNESS IN CLASSIFICATION

We address fairness at two complementary levels: *prediction* and *representation*. At the **prediction level**, fairness requires making model outputs independent of the sensitive attribute $S$. For data-label pair $(X, Y)$, a classifier $f : X \to \hat{Y}$ is considered fair if $P(\hat{Y} \mid S = s) = P(\hat{Y} \mid S = s')$, i.e., changes in $S$ should not affect the predicted label. This notion of fairness has been widely studied in prior work, including Park et al. (2022); Tian et al. (2024), which analyze prediction-level fairness criteria in vision models. At the **representation level**, fairness enforces that learned embeddings $e(X)$ do not encode sensitive information: $I(e(X); S) = 0$, where $I(\cdot \, ; \cdot)$ denotes mutual information. This ensures that sensitive attributes cannot be inferred from internal representations. Representative approaches targeting this form of fairness include Ravfogel et al. (2020); Kumar et al. (2023). Together, these two levels encourage models that both avoid biased predictions and suppress sensitive information in embedding space. Next, we outline how our model supports these objectives.

## 3 FAIRNVT FRAMEWORK

FairNVT mitigates bias in pre-trained embeddings and promotes fair predictions by injecting calibrated noise and optimizing with fairness-aware objectives. Figure 1 shows an overview of the proposed framework. Specifically, we attach light-weight adapters to extract task-relevant and sensitive information from the potentially biased embedding. Then, the sensitive embedding is perturbed with random, calibrated Gaussian noise and fused with the task embedding for downstream classification. The model is jointly optimized with classification, orthogonality, and demographic parity losses to balance multiple objectives. Section 3.1 discusses the key components of the model. Section 3.2 introduces the optimization objectives, and 3.3 describes the training and inference procedures. Appendix B presents the mathematical intuition between connecting prediction- and representation-level fairness that motivates the design of the framework.

### 3.1 MODEL COMPONENTS

**Adapters and classification heads.** We use the Adapter modules to extract task-relevant and sensitive information from the frozen pre-trained models, with supervision from the task and sensitive

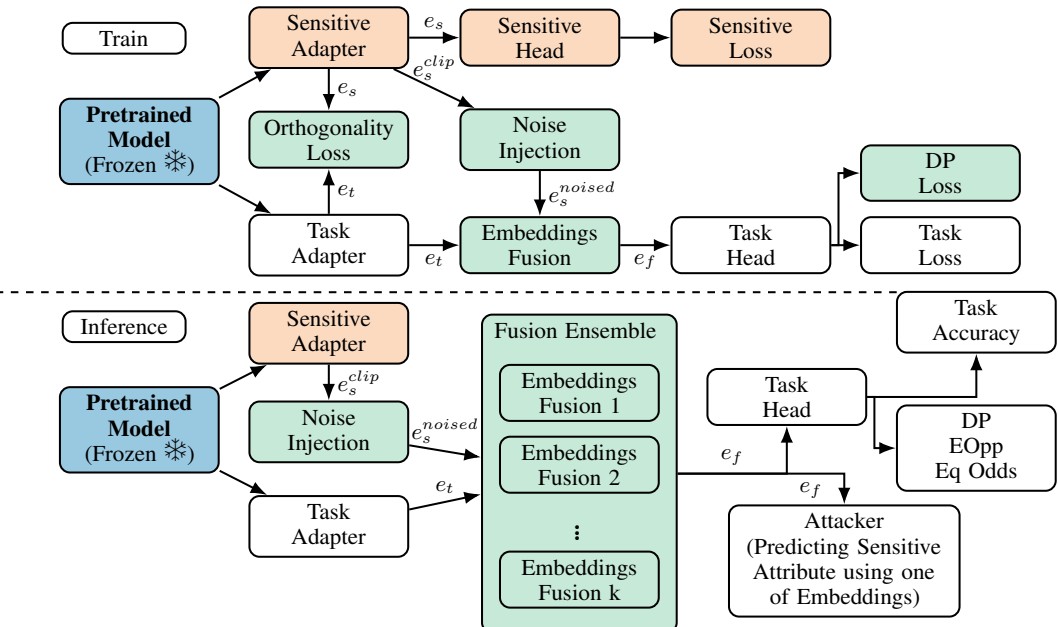

Figure 1: **Overview of the proposed FairNVT framework.** During *training*, a frozen ViT backbone is attached with lightweight task and sensitive adapters. The adapters yield task ($e_t$) and sensitive ($e_s$) embeddings. The sensitive path inputs $e_s$ for the sensitive head and a clipped and noised embedding $e_s^{\text{noised}}$ ($e_s^{\text{clip}}$ injected with noise), that is concatenated with $e_t$ to get the fused embedding $e_f$ for task prediction. We jointly optimize a weighted sum of task and sensitive classification losses, orthogonality and demographic-parity losses. During *inference*, multiple noise samples produce a fused embedding $e_f$ whose task predictions are aggregated by majority vote. Blue marks the frozen backbone; Orange extracts sensitive information; Green performs debiasing, fusion, and prediction.

labels. The Adapters are model-agnostic, lightweight blocks of trainable parameters attached to various blocks of the frozen pre-trained model. For example, for the image and text classification tasks which we use the Vision Transformer (ViT Dosovitskiy et al. (2021)) and BERT-Base Turc et al. (2019) models as frozen pre-trained models, the Adapters [1] are bottleneck feed-forward layers attached to each Transformer block, consisting of down-projection matrix to project the hidden states into a lower dimension layer, and an up-projection matrix to project back into the original hidden dimension (Poth et al., 2023). To encourage disentangling task and sensitive embeddings, we attach separate Adapters and classification heads to learn the task and sensitive labels respectively. We use the class token representation (`[CLS]` token embedding) from the adapted ViT and BERT model as the task ($e_t$) and sensitive ($e_s$) embeddings, with only the task and sensitive Adapters activated respectively. The classification heads are simple Multi-layer Perceptrons (MLPs) that takes in the adapted embeddings and predict the task and sensitive labels respectively.

**Noise injection.** To introduce perturbation to the sensitive information, we clip the sensitive embedding and add random noise sampled from a Gaussian distribution. Specifically, let $e_s$ be the sensitive embedding vector, we clip the sensitive embedding to upper-bound its $L2$-norm to $C$, $e_s^{\text{clip}} = e_s / \max(1, \frac{\|e_s\|_2}{C})$, where $C$ is a hyperparameter to control the embedding scale. The clipping procedure ensures that we calibrate the noise level to the scale of the embedding, thus better controlling the strength of perturbations. The noise $z$ is then randomly drawn from a Normal distribution with mean equals zero, such that the perturbed embedding is unbiased, and with the variance scales with $C$, i.e., $z \sim \mathcal{N}(0, C^2\sigma^2\mathbb{I}^d)$, where $\sigma$ controls the noise level and $d$ is the dimension of $e_s$. Finally, the perturbed embedding is obtained by adding noise $z$ isotropically to the sensitive embedding, $e_s^{\text{noised}} = e_s^{\text{clip}} + z$.

Injecting calibrated noise has two effects in terms of making fair task predictions. First, it perturbs the sensitive information extracted from the sensitive adapter, such that the model learns to depend less on

---

[1]https://docs.adapterhub.ml/methods.html#bottleneck-adapters

irrelevant sensitive information when predicting the task label. Second, the task information extracted from the task adapter is preserved but perturbed with noisy sensitive information, to encourage more robust predictions of the task labels from the noisy embedding.

**Embedding fusion.** After obtaining the perturbed sensitive embedding, we concatenate it with the task embedding, and use it as a de-biased representation for downstream classification tasks. We demonstrate that simple concatenations can effectively preserve task-relevant information for making fair predictions, while obfuscating sensitive information such that sensitive labels cannot be easily inferred.

We add two steps to ensure that the task and sensitive Adapter capturing the correct information such that the fused embedding works as expected. The noised sensitive embedding only enters the task label classification head and not the sensitive head, such that the sensitive adapter captures the clean sensitive information. The gradient is blocked from the task loss to the noised sensitive embedding, such that the learned task information does not interfere with the sensitive Adapter. We provide a short explanation in supplementary materials, illustrating why noise applied in the sensitive subspace reduces the amount of sensitive information available to adversarial predictors.

## 3.2 OPTIMIZATION OBJECTIVES

The proposed framework is trained jointly with classification, orthogonality and demographic parity losses to balance between making accurate task predictions and maintaining fairness with respect to the sensitive information. We describe each loss and their objectives in this section.

**Classification loss.** The cross-entropy loss is used for each classification head to evaluate the predictive performances. Let $i, k$ be the sample and class index, $\theta$ be the model parameters, $(x_i, y_i)$ be each data-label pair, $\hat{y}$ be the predicted label, then the cross-entropy loss for predicting task ($t$) and sensitive ($s$) labels are,

$$L_{\text{ce}}^{\alpha}(\theta) = -\frac{1}{n} \sum_{i=1}^{n} \sum_{k=1}^{K} y_{i,\alpha}^{k} \log p_{\theta}(\hat{y}_{i,\alpha} = k | x_i), \alpha \in \{s, t\}. \tag{1}$$

**Orthogonality loss.** It is common that sensitive information might help predicting the task label as they share common features. For example, when the task is predicting 'wearing glasses' and the sensitive label is 'Age', the features might be correlated as it might coincide that elder people often wear glasses. In such cases, we want to penalize similar patterns in the task and sensitive embeddings and encourage finding distinct features in predicting the task label. We use mean cosine similarity to quantify the similarities between task and sensitive embeddings, and penalize for higher scores to encourage de-correlating these embeddings. Let $e_{t,i}, e_{s,i}$ denote the per-sample task and sensitive embedding that depend on $\theta$,

$$L_{\text{orth}}(\theta) = \frac{1}{n} \sum_{i=1}^{n} \left( \frac{e_{t,i}^{\top} e_{s,i}}{||e_{t,i}||_2 ||e_{s,i}||_2} \right)^2. \tag{2}$$

**Demographic parity loss.** While the orthogonality loss and noise injection help disentangling the sensitive information, to ensure making fair predictions, we add a demographic parity loss to encourage learning similar logits among different sensitive groups. Following the definition of demographic parity difference Agarwal et al. (2019),

let $n_0, n_1$ be the number of samples in a batch belonging to sensitive group $0, 1$ respectively, $p_i = p_{\theta}(\hat{y}_i = 1 | x_i)$ be the probability of predicting the positive class of $y$, and $\mathbf{1}[\cdot]$ be the indicator function then,

$$L_{\text{dp}}(\theta) = \left| \frac{1}{n_0} \sum_{i=1}^{n_0} \mathbf{1}[s_i = 0] p_i - \frac{1}{n_1} \sum_{i=1}^{n_1} \mathbf{1}[s_i = 1] p_i \right|. \tag{3}$$

The overall loss is weighted to adjust the scale differences between the three losses and to allow flexibility of viewing different importance of the optimization targets,

$$L = L_{\text{ce}}^{\text{t}} + \beta_1 L_{\text{ce}}^{\text{s}} + \beta_2 L_{\text{orth}} + \beta_3 L_{\text{dp}}, \tag{4}$$

where $\beta$s are hyperparameters representing weights on each loss.

### 3.3 TRAINING AND INFERENCE PROCEDURES

The arrows in Figure 1 shows the forward pass direction. In the backward pass, only the Adapters and classification heads parameters are updated with loss $L$, while the pre-trained model remains frozen. The noise injection and embedding concatenation steps do not induce learnable parameters. In the training stage, we optimize for a single draw of noise to predict the true task label. At the inference stage, multiple samples of noise are drawn from the same distribution as optimized in training, to enable majority voting to improve test accuracy, and to quantify fair certifications (Appendix B). The sensitive label is only used for calculating the demographic parity loss in the training stage and not accessed during inferences, as the classifiers remain frozen at the trained parameters.

## 4 EXPERIMENTS

### 4.1 EXPERIMENT SETUPS

**Implementation and hyperparameters.** For both our method and the baselines, we train the models using AdamW Loshchilov & Hutter (2019); Kingma & Ba (2015) with batch size 256 and default hyper parameters. The adapter architecture uses a reduction factor of 8 for the task branch and 16 for the sensitive branch. We run a grid search over other sensitive hyperparameters including learning rates and loss weights, and report the best validation-selected results. Full hyperparameter ranges, sensitivity analyses, and compute details are provided in the supplementary materials.

**Datasets[2] and tasks.** We evaluate **image-based classification tasks** with ViT as the frozen backbone to compare with state of the art baselines such as Tian et al. (2024). We use the publicly available CelebA Liu et al. (2015) and UTKFace Zhang et al. (2017) datasets for facial attribute classification. CelebA contains roughly 200K images with attribute annotations. Following prior work Tian et al. (2024); Park et al. (2022), we consider perceived gender or age as sensitive attributes, and we study target attributes such as expression(smiling), big nose and wavy hair. We use the official train/validation/test splits. UTKFace contains approximately 20K images with annotations including gender and age. To follow a binary fairness formulation Park et al. (2022), we group age into $< 35$ vs. $\geq 35$ and use age as the sensitive attribute and gender as the target attribute. The dataset is divided into three subsets, where subsets 1, 2, and 3 are used for training, validation, and testing, respectively.

We additionally evaluate on **text-based classification tasks** with Bert-Base as the frozen backbone to compare with baselines Kumar et al. (2023); Ravfogel et al. (2020). The dataset BIOS De-Arteaga et al. (2019) consists of professional biographies with occupation labels, and the task is to predict occupation while evaluating fairness with respect to perceived gender.

**Metrics.** We evaluate task performance, prediction-level and representation-level fairness performances with standard metrics[3] used by baseline methods.

(1) Task performance is measured using *accuracy* (Acc) and *balanced accuracy* (BAcc), where BAcc accounts for label imbalance across classes. Given true labels $Y \in \{0, 1\}$ and predictions $\hat{Y}$,

$$BAcc \coloneqq \frac{1}{2}\left(\frac{\text{TP}}{\text{TP} + \text{FN}} + \frac{\text{TN}}{\text{TN} + \text{FP}}\right), \tag{5}$$

where TP, TN, FP, and FN denote true/false positives/negatives, respectively.

To assess prediction-level fairness, we employ three widely used group fairness metrics: Let $S \in \{0, 1\}$ denote the binary sensitive attribute, the metrics are as follows:

(2) *Demographic Parity (DP)* computes the difference between the largest and smallest rates across all groups:

$$DP \coloneqq \max_s \mathbb{E}[\hat{Y} \mid S] - \min_s \mathbb{E}[\hat{Y} \mid S], \tag{6}$$

---

[2]The datasets are publicly available and include perceived annotations provided by the dataset creators. We use these labels solely for modeling and fairness evaluation to compare with previously published results on these benchmarks.

[3]https://fairlearn.org/

which simplifies to $|\mathbb{E}[\hat{Y} = 1 \mid S = 0] - \mathbb{E}[\hat{Y} = 1 \mid S = 1]|$ in the binary case.

(3) *Equalized Odds (EO)* adds conditioning on the task label compared to *DP*:

$$EO := \frac{1}{2}\sum_{y \in \{0,1\}} \Big|\mathbb{E}[\hat{Y} = 1 \mid Y = y, S = 0] - \mathbb{E}[\hat{Y} = 1 \mid Y = y, S = 1]\Big|, \qquad (7)$$

(4) *Equal Opportunity (EOpp)* is a relaxed version of *EO* that only considers conditional expectations with respect to positive task labels:

$$EOpp := |\mathbb{E}[\hat{Y} = 1 \mid Y = 1, S = 0] - \mathbb{E}[\hat{Y} = 1 \mid Y = 1, S = 1]|. \qquad (8)$$

We report absolute DP/EO/EOpp gaps throughout, in all three metrics, lower values indicate higher fairness level.

To assess representation-level fairness, we examine the prediction accuracy of the sensitive attribute from an attacker. Lower values indicate higher fairness level.

(5) *Attacker accuracy (Att.Acc)* measures sensitive-information leakage using a post-hoc attacker: a MLP trained to predict $S$ from embeddings ($e_f$) at a saved checkpoint (encoder frozen). Lower Att.Acc indicates less recoverable sensitive information and thus stronger representation-level fairness. Architecture/training details and an ablation on attacker depth are provided in the supplementary materials.

## 4.2 BASELINES

We compare our approach with representative fairness methods under a unified evaluation protocol.

**Image-based Classification.** We include three baselines[4]:

- **Vanilla (ViT)** Dosovitskiy et al. (2021): ViT with a task adapter and classification head trained; no fairness intervention.
- **ViT-FSCL** Park et al. (2022): Representation-level contrastive debiasing; we re-implement it on a ViT backbone for consistent comparison.
- **FairViT** Tian et al. (2024): Architecture-level debiasing via adaptive masking on ViT attention maps.

**Text-based Classification** We include five baselines:

- **Vanilla-BERT** Devlin et al. (2019): Standard fine-tuning without fairness intervention.
- **FT-Debias** Kumar et al. (2023): Fine-tuning with adversarial debiasing objectives.
- **INLP** Ravfogel et al. (2020): Iteratively trains linear probes on the sensitive attribute and projects embeddings to remove the corresponding subspaces.
- **SUP** Shi et al. (2024): Projection-based concept removal that preserves task-relevant features while suppressing sensitive directions.
- **ConGater** Masoudian et al. (2024): Group-aware contrastive training to disentangle task and sensitive representations.
- **DAM** Kumar et al. (2023): Parameter-efficient debiasing using adapter fusion to reduce demographic leakage.

## 4.3 EXPERIMENT RESULTS

We compare our proposed FairNVT with several baselines on CelebA, UTKFace and BIOS datasets in Table 1 and 2 respectively. The bold and underlined numbers indicate the best and second best results. We tune the hyperparameters for all methods based on the highest task accuracy and report the mean values and standard deviations over 3 runs. Best results are shown in **bold**, and the second-best results are underlined. Overall, we observe that *FairNVT achieves significantly lower attacker accuracy and fairer downstream predictions, while preserving strong task performance.*

---

[4]FairVPT Park & Byun (2024) is not included due to the lack of an official implementation at submission time.

Table 1: **Image-Based Classification task:** Comparing our method with baselines on CelebA (a-c) and UTKFace (d) dataset. FairNVT demonstrates strong performance in higher task performance while achieving fairer predictions.

| Method | Acc(↑) | BAcc(↑) | DP(↓) | EOpp(↓) | EO(↓) | Att.Acc(↓) |
|---|---|---|---|---|---|---|
| Vanilla | 89.6±0.1 | 89.0±0.1 | 16.9±0.2 | 8.4±1.2 | 6.6±1.2 | 98.7±0.0 |
| ViT-FSCL | 89.9±1.0 | 87.1±1.0 | 14.5±2.2 | 6.9±2.2 | 5.1±2.1 | 97.7±0.1 |
| FairViT | 92.7±0.2 | 92.0±0.3 | 16.0±0.3 | 4.3±0.4 | 2.7±0.6 | 97.0±0.1 |
| FairVPT | 91.6±0.2 | 91.4±0.2 | 13.9±0.3 | 2.4±0.3 | 1.8±0.6 | 98.6±0.2 |
| FairNVT(Ours) | 93.1±0.2 | 93.0±0.3 | 9.9±0.3 | 0.8±0.3 | 1.5±0.5 | 51.6±0.4 |

(a) Task: Expression (Smiling), Sensitive Attribute: Gender (Male)

| Method | Acc(↑) | BAcc(↑) | DP(↓) | EOpp(↓) | EO(↓) | Att.Acc(↓) |
|---|---|---|---|---|---|---|
| Vanilla | 80.2±0.2 | 63.4±1.2 | 25.0±1.2 | 19.2±0.2 | 25.2±1.2 | 88.3±0.0 |
| ViT-FSCL | 81.5±0.8 | 68.1±1.6 | 25.5±4.4 | 23.2±4.2 | 18.3±3.8 | 87.0±0.3 |
| FairViT | 84.6±0.2 | 69.9±0.2 | 22.7±0.8 | 22.9±1.4 | 16.7±1.2 | 86.0±0.3 |
| FairVPT | 83.1±0.2 | 64.0±0.3 | 17.8±1.0 | 23.0±1.0 | 15.1±0.5 | 87.5±0.2 |
| FairNVT(Ours) | 82.1±0.2 | 69.2±0.5 | 10.9±1.5 | 2.3±0.8 | 1.9±0.2 | 67.6±0.6 |

(b) Task: Big Nose, Sensitive Attribute: Age (Young)

| Method | Acc(↑) | BAcc(↑) | DP(↓) | EOpp(↓) | EO(↓) | Att.Acc(↓) |
|---|---|---|---|---|---|---|
| Vanilla | 84.4±0.5 | 76.1±0.1 | 31.1±2.7 | 31.3±2.5 | 34.9±2.7 | 98.6±0.1 |
| ViT-FSCL | 83.5±0.5 | 69.8±2.7 | 31.0±1.1 | 28.9±4.1 | 36.7±11.6 | 97.6±0.1 |
| FairViT | 86.4±0.4 | 79.9±0.3 | 38.0±0.8 | 30.0±1.1 | 20.9±0.9 | 94.2±0.2 |
| FairVPT | 84.6±0.4 | 76.5±0.4 | 31.8±0.8 | 28.5±0.8 | 17.5±1.0 | 98.4±0.1 |
| FairNVT (Ours) | 84.7±0.3 | 82.3±0.2 | 18.9±0.9 | 5.6±0.6 | 6.3±0.7 | 62.8±0.4 |

(c) Task: Wavy Hair, Sensitive Attribute: Gender (Male)

| Method | Acc(↑) | BAcc(↑) | DP(↓) | EOpp(↓) | EO(↓) | Att.Acc(↓) |
|---|---|---|---|---|---|---|
| Vanilla | 97.3±0.1 | 96.0±0.5 | 19.5±0.3 | 1.3±0.2 | 3.1±0.2 | 82.7±0.5 |
| ViT-FSCL | 97.4±0.2 | 96.7±0.5 | 19.2±0.1 | 2.2±0.3 | 1.1±0.1 | 82.3±0.2 |
| FairViT | 97.5±0.0 | 97.1±0.1 | 21.0±0.9 | 1.8±0.3 | 1.1±0.4 | 81.0±0.2 |
| FairVPT | 95.3±0.1 | 93.9±0.2 | 19.4±0.2 | 2.0±0.4 | 2.0±0.5 | 74.1±0.2 |
| FairNVT(Ours) | 97.7±0.5 | 97.4±0.5 | 18.4±0.7 | 0.6±0.2 | 1.5±0.7 | 50.2±1.0 |

(d) Task: Gender, Sensitive Attribute: Age

**Comparison on CelebA.** As shown in Table 1 (a-c), FairNVT maintains a balanced trade-off between fairness and task performance across all task–sensitive pairs. Our method achieves comparable or higher task accuracies while consistently improving multiple fairness metrics. Notably, it reduces sensitive information leakage as indicated by attacker accuracy dropping by 30% and close to random, demonstrating the effectiveness of the proposed noise-based regularization. We observe that our method reduces both metrics simultaneously, suggesting a reduced dependence on the sensitive attribute at the representation level.

**Comparison on UTKFace.** Table 1 (d) presents the results on the UTKFace Zhang et al. (2017) dataset where *Gender* and *Age* are treated as task and Sensitive attribute respectively. FairNVT achieves consistently strong fairness improvements while maintaining comparable task accuracy, achieving the best balanced accuracy and reducing fairness disparity across most metrics.

**Comparison on BIOS.** Table 2 reports results on BIOS De-Arteaga et al. (2019), where the task is multi-class *Profession*[5] and the sensitive attribute is *Gender*. Overall, FairNVT offers a favorable fairness–utility trade-off, pairing competitive accuracy with state-of-the-art DP and sensitive-attribute leakage close to the best baseline. Figure 2 compares logits for original and gender-swapped sentences, where pronouns are replaced with those of the opposite gender. FairNVT produces more similar distributions between the two, indicating reduced sensitivity to gender. Additional details, illustrative examples, and predicted scores are provided in the supplementary materials. These results highlight that FairNVT can effectively extend from the vision domain to textual embeddings.

**Qualitative results.** Figure 4 visualizes model attributions on *CelebA*, where task is *expression (smiling)* and sensitive attribute is *gender (male)*. We observe that the Vanilla model frequently relies on gender-correlated regions (e.g., hair/beard), while ViT-FSCL and FairViT partially down-weight such cues. In contrast, FairNVT consistently attends more to expression-relevant regions such as the mouth, cheeks, and eyes, while effectively suppressing gender-related cues. This behavior aligns with the observed quantitative improvements in DP and EOpp, as well as the significant reduction in attacker accuracy.

## 4.4 ABLATION STUDIES

We choose the *expression (smiling)* task and *gender (male)* sensitive attribute for ablation studies.

**Effect of different model components.** Table 3 presents the ablation study evaluating the contribution of different components in the proposed method. Across settings, task accuracy and balanced

---

[5]EO and EOpp condition on a binary label and are not directly applicable to multi-class tasks; DP remains applicable.

Table 2: **Text-Based Classification task:** Comparing our method with baselines on Bios De-Arteaga et al. (2019) dataset, Task: Profession (Multi-Class), Sensitive Attribute: Gender. All reported values are scaled by $\times 10^2$.

| Method | Acc($\uparrow$) | DP($\downarrow$) | Att.Acc($\downarrow$) |
|---|---|---|---|
| **Vanilla-BERT** | $72.8_{\pm 0.2}$ | $2.0_{\pm 0.2}$ | $99.6_{\pm 0.0}$ |
| **+FT-Debias** | $76.8_{\pm 2.4}$ | $2.1_{\pm 0.2}$ | $58.4_{\pm 0.3}$ |
| **+INLP** | $76.4_{\pm 0.1}$ | $\underline{1.7}_{\pm 0.1}$ | $\mathbf{51.9}_{\pm 0.2}$ |
| **+SUP** | $77.2_{\pm 0.2}$ | $2.1_{\pm 0.5}$ | $74.3_{\pm 0.6}$ |
| **+DAM** | $80.3_{\pm 0.4}$ | $2.2_{\pm 0.5}$ | $60.6_{\pm 0.2}$ |
| **+CONGATER** | $\mathbf{82.4}_{\pm 0.5}$ | $1.9_{\pm 0.3}$ | $59.0_{\pm 0.2}$ |
| **+FairNVT(Ours)** | $\underline{80.6}_{\pm 0.4}$ | $\mathbf{1.6}_{\pm 0.1}$ | $\underline{52.8}_{\pm 0.3}$ |

Figure 2: **Robustness to gender-indicator swapping on BIOS.** We plot the distribution of the model's confidence in predicting profession for the original text and its gender-swapped counterpart for 100 random samples. FairNVT (right) exhibits more overlapping distributions than Vanilla (left) in more confident predictions.

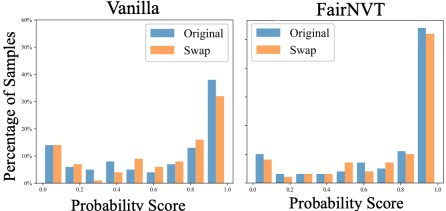

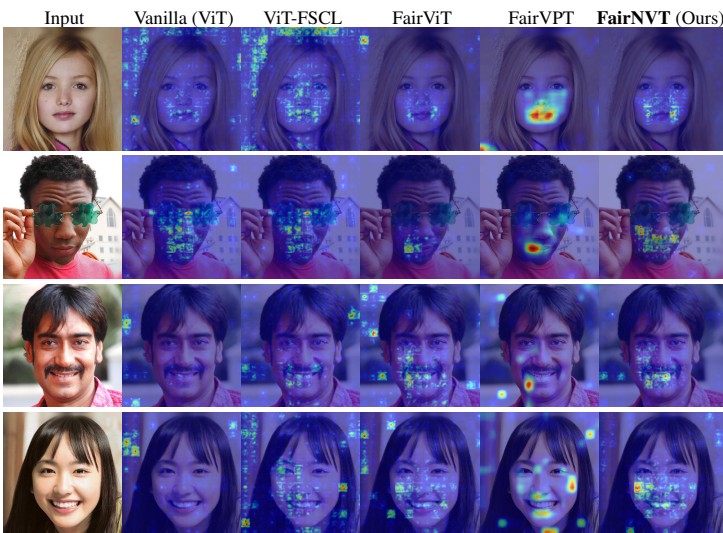

Figure 3: **Gradient-based saliency map** for the *Expression (smiling)* as main task and *Gender (male)* as sensitive attribute. Warmer regions indicate stronger contribution to the output logit. FairNVT primarily attends to expression-relevant areas (mouth/cheeks), demonstrating reduced reliance on gender-correlated cues.

accuracy remain broadly stable, indicating that these fairness components do not degrade utility. Splitting into task and sensitive adapters already improves fairness metrics from the Vanilla baseline, suggesting that disentangling information with supervision from both task and sensitive labels is more effective than learning from the task label directly. The orthogonality loss encourages the task

Table 3: **Ablation of FairNVT components on CelebA.** We toggle Demographic Parity loss (DP), Orthogonality loss (Orth), and Noise injection (Noise) for *expression (smiling)* as main task and *gender (male)* as sensitive attribute. ✓and ✗means the component is present and absent respectively. DP loss consistently drives fairness, Noise suppresses sensitive attribute leakage, and Orth further improves fairness, with minimal utility change.

| Fair Loss | Orth Loss | Noise | Acc($\uparrow$) | BAcc($\uparrow$) | DP($\downarrow$) | EOpp($\downarrow$) | EO($\downarrow$) | Att Acc($\downarrow$) |
|---|---|---|---|---|---|---|---|---|
| ✗ | ✗ | ✗ | $92.7_{\pm 0.1}$ | $92.2_{\pm 0.1}$ | $13.8_{\pm 0.6}$ | $4.8_{\pm 0.4}$ | $2.8_{\pm 0.5}$ | $98.4_{\pm 0.0}$ |
| ✗ | ✓ | ✓ | $\mathbf{93.2}_{\pm 0.1}$ | $92.8_{\pm 0.1}$ | $14.6_{\pm 0.5}$ | $4.9_{\pm 0.5}$ | $\underline{2.2}_{\pm 0.6}$ | $\underline{52.8}_{\pm 0.5}$ |
| ✓ | ✗ | ✓ | $92.6_{\pm 0.1}$ | $92.8_{\pm 0.1}$ | $\mathbf{9.9}_{\pm 0.4}$ | $\underline{1.1}_{\pm 0.4}$ | $2.4_{\pm 0.5}$ | $53.0_{\pm 0.3}$ |
| ✓ | ✓ | ✗ | $92.9_{\pm 0.3}$ | $92.9_{\pm 0.2}$ | $\underline{10.1}_{\pm 0.4}$ | $2.4_{\pm 0.4}$ | $3.0_{\pm 0.4}$ | $98.5_{\pm 0.1}$ |
| ✓ | ✓ | ✓ | $\underline{93.1}_{\pm 0.2}$ | $\mathbf{93.0}_{\pm 0.3}$ | $\mathbf{9.9}_{\pm 0.3}$ | $\mathbf{0.8}_{\pm 0.3}$ | $\mathbf{1.5}_{\pm 0.5}$ | $\mathbf{51.6}_{\pm 0.4}$ |

Table 4: **Sensitivity to noise.** We ablate on noise levels for *expression (smiling)* as main task and *gender (male)* as sensitive attribute. Moderate noise levels balance utility (Acc/BAcc), fairness gaps (DP/EOpp/EO), and sensitive information leakage (Att. Acc). Very large noise further improves most fairness metrics but begins to slightly reduce accuracy, reflecting a utility-fairness trade-off at higher noise levels.

| Noise Level ($\sigma$) | Acc($\uparrow$) | BAcc($\uparrow$) | DP($\downarrow$) | EOpp($\downarrow$) | EO($\downarrow$) | Att.Acc($\downarrow$) |
|---|---|---|---|---|---|---|
| 1 | $93.0_{\pm 0.2}$ | $\mathbf{93.1}_{\pm \mathbf{0.2}}$ | $9.4_{\pm 0.4}$ | $1.0_{\pm 0.2}$ | $2.0_{\pm 0.4}$ | $67.4_{\pm 0.2}$ |
| 5 | $\mathbf{93.1}_{\pm \mathbf{0.2}}$ | $93.0_{\pm 0.3}$ | $9.9_{\pm 0.3}$ | $\mathbf{0.8}_{\pm \mathbf{0.3}}$ | $1.5_{\pm 0.5}$ | $51.6_{\pm 0.4}$ |
| 100 | $91.0_{\pm 0.3}$ | $91.2_{\pm 0.2}$ | $\mathbf{9.2}_{\pm \mathbf{0.5}}$ | $0.9_{\pm 0.5}$ | $\mathbf{1.1}_{\pm \mathbf{0.4}}$ | $\mathbf{50.5}_{\pm \mathbf{0.3}}$ |

and sensitive adapters to capture distinct features; Without it, the model can mix task-relevant and sensitive signals, and may inadvertently discard information that is also predictive for the task during debiasing. The DP loss consistently strengthens fairness, especially DP and EOpp, by explicitly reducing dependence between predictions and the sensitive attribute. Finally, noise is critical for leakage reduction: removing it weakens all fairness measures, with the clearest impact on attacker accuracy, confirming that perturbing the sensitive adapter embedding effectively conceals sensitive attributes.

**Analysis of noise strength ($\sigma$).** We analyze how the noise strength $\sigma$ influences both representation and prediction level fairness. As shown in Table 4, moderate noise substantially lowers attacker accuracy, indicating that the injected perturbation effectively hides sensitive information without disturbing the task signal. When the noise becomes very large, the model shows improvements in several fairness metrics (DP, EO, Att.Acc) but shows a slight decline in predictive accuracy. In practice, a moderate noise level provides a stable trade-off between privacy and utility. Additional experiments related to component isolation (noise only, DP loss only, orthogonality loss only) and sensitivity analyses of the corresponding loss weights are included in the supplementary materials.

## 5 CONCLUSION

We introduced FairNVT, a plug-in framework that injects calibrated Gaussian noise in a learned sensitive subspace to improve both representation and prediction level fairness . Across image and text tasks, it consistently reduces sensitive-attribute leakage and matches or improves accuracy. While our study focuses on image and text modalities, the same recipe naturally extends to additional modalities and a wide range of transformer-encoder–based architectures. We are excited about these directions and expect the approach to scale with little engineering overhead, motivating researchers to broaden applicability beyond text and image to new modalities and transformer-based models.

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

## A  RELATED WORKS

**Transformers for Classification.**   Transformers have seen broad adoption for classification in both vision and text. Image-level transformer models such as ViT Dosovitskiy et al. (2021) have been widely used across domains including face analysis Dan et al. (2023); Narayan et al. (2025); Jacob & Stenger (2021), medical imaging Shao et al. (2021); Tang et al. (2022), and general object recognition Khan et al. (2022); Wang et al. (2025). These backbones match or surpass strong CNN models while offering flexible transfer to new datasets.

Similarly, transformer-based language models (e.g., BERT Devlin et al. (2019), RoBERTa Liu et al. (2019), DeBERTa He et al. (2021)) have become the dominant choice for text classification, often outperforming CNN/RNN architectures and transferring effectively via pretrain–adapt pipelines.

Given their importance, understanding and mitigating their fairness challenges is crucial. We focus on image classification with frozen vision transformers and show that the proposed framework also transfers effectively to text.

**Fairness Approaches.**   Many approaches mitigate group disparities by modifying the training distribution itself. Classical methods include reweighing Kamiran & Calders (2012), disparate impact removal Feldman et al. (2015), and optimized preprocessing Calmon et al. (2017), which explicitly adjust sample weights or features to balance sensitive groups. More recent strategies alter the training data more subtly through curated fine-tuning Ghanbarzadeh et al. (2023), group rebalancing, or fairness-oriented augmentation Sun et al. (2023); Halevy et al. (2025), aiming to reduce distributional bias without modifying model parameters.

Unlike data modification approaches, we make no data level changes, group labels are used only at training to optimize demographic parity, and not required at inference.

Beyond data manipulation, fairness has also been pursued through changing the learning objective Agarwal et al. (2018); Zhang et al. (2018). More recently, transformer-based methods adjust attention Zhou et al. (2024), mask bias-correlated ViT regions Tian et al. (2024), or apply fairness-aware prompting Park & Byun (2024). These recent approaches typically adjust attention or prompting, whereas we operate in a learned sensitive latent subspace and apply randomized smoothing without architectural changes or retraining the frozen model.

Another direction introduces parameter-efficient modules for debiasing, such as adapters Fatemi et al. (2023); Hauzenberger et al. (2023); Yang et al. (2023); Lauscher et al. (2021); Kumar et al. (2023); Masoudian et al. (2024). For example, DAM Kumar et al. (2023) adds debiasing adapters alongside task adapters to handle multiple sensitive attributes, while ConGater Masoudian et al. (2024) introduces controllable gates that balance fairness and utility at inference time. Although these approaches lower training cost, they typically act indirectly on representations without explicitly identifying or perturbing a sensitive subspace. In contrast, our method keeps the transformer backbone frozen and directly manipulates a learned sensitive subspace through noise injection.

Recent methods improve fairness by directly altering latent representations, with approaches based on latent factorization or variational modeling Zemel et al. (2013); Louizos et al. (2015) and adversarially aligned representations Madras et al. (2018); Zhang et al. (2018); Götte (2023) that aim to reduce sensitive information in learned features through min–max optimization, can be unstable and often requiring multi-stage training.

Projection-based methods such as INLP Ravfogel et al. (2020), sufficient projection (SUP) Shi et al. (2024), and SLSD Islam et al. (2024) remove subspaces predictive of sensitive attributes; however, linear removal can discard task-relevant information when sensitive and semantic directions overlap. Information-theoretic approaches Kang et al. (2022); Wang et al. (2023); Xie et al. (2024) estimate mutual information to regularize fairness, while contrastive debiasing Park et al. (2022); Shen et al. (2021) often relies on group-balanced sampling and two-stage optimization. Recent concept-editing methods Karvonen et al. (2024) learn sparse subspaces aligned with sensitive concepts and suppress them to reduce probe recoverability.

**Fairness via Randomized Smoothing.** Randomized smoothing Lecuyer et al. (2019); Cohen et al. (2019) is primarily studied as a robustness technique, where prediction stability under noise yields certified guarantees for robust predictions. Although not originally developed for fairness, the resulting invariance suggests that smoothing could help reducing reliance on sensitive factors.

Individual fairness is formalized via task-relevant similarity metrics Dwork et al. (2012). Empirical work connecting smoothing to fairness remains limited. For example, Jin et al. (2022) trains group-specific models and averages their parameters to certify group fairness in low-dimensional tabular settings, while Yeom & Fredrikson (2021); Peychev et al. (2022) encourage individual fairness via smoothing in input or latent spaces. These approaches, however, require isolated sensitive attributes in tabular data style, or operate only when input perturbations are well defined. Unlike prior works, our method performs smoothing selectively in a learned sensitive subspace, suppressing sensitive variation while preserving task structure, without architectural changes or using sensitive labels at inference.

## B  OBFUSCATING SENSITIVE INFORMATION IMPROVES FAIRNESS

In this section, we explain the mathematical intuition to the design of the FairNVT framework. The goal of achieving prediction-level fairness, as measured by Demographic Parity (DP) and Equalized Odds (EO), is to ensure the model predictions are similar across different sensitive groups, i.e. $P(\hat{Y}|S=0) = P(\hat{Y}|S=1)$ for DP, and $P(\hat{Y}|S=0, Y=y) = P(\hat{Y}|S=1, Y=y)$ for EO. In typical debiasing pipelines, the predictions often depend on both the representation $Z$ learnt from data $X$ and the sensitive information, i.e. $\hat{Y} = f(Z = e(X), S)$. This motivates the use of fair representations that suppress sensitive information in the learned embedding, resulting in predictions of the form $\hat{Y} = f(Z = e(X)), Z \perp\!\!\!\perp S$. By limiting the model's access to sensitive attributes, such representation removes unconditional dependence on $S$, and encourage alignment of conditional prediction behavior across groups, thereby reducing disparities in both DP and EO.

We formalize the intuition that noising sensitive information in the task classifier embedding improves both prediction- and representation-level fairness. Let $(X, Y, S)$ be the data pair that represents features, task and sensitive attributes respectively. Let $Z = e(X)$ be the encoded embedding of $X$ from an encoder model $e$ (e.g. the frozen backbone models). Given a realized embedding $z$, let $c$ be a classifier model that predicts task attribute $\hat{Y}$ with $\mathbb{1}(c(z) > \tau)$ where given a threshold $\tau$. In the case where both $S, Y$ are binary attributes, the following result follows directly from the definition of total variation distance between two probability measures.

**Lemma B.1.** *If $Z \perp\!\!\!\perp S$, then $DP = 0$, $EO = 0$, $EOpp = 0$.*

*Proof.* Let $A$ be the event that the classifier $c$ predicts $\hat{Y} = 1$, i.e. $A = \{z : \mathbb{1}(c(z) > \tau) = 1\}$, and let $P, Q$ be the conditional distribution of $Z|S = 0$ and $Z|S = 1$ where $P(A) = \Pr(\hat{Y} = 1|S = 0)$, $Q(A) = \Pr(\hat{Y} = 1|S = 1)$, then by the definition of demographic parity difference ($DP$) and total variation distance ($\delta_{TV}$),

$$DP := |P(\hat{Y} = 1|S = 0) - P(\hat{Y} = 1|S = 1)| = |P(A) - Q(A)| \leq \sup_A |P(A) - Q(A)| := \delta_{TV}(P, Q).$$

If $Z \perp\!\!\!\perp S$, then $P = Q$ and $\delta_{TV}(P, Q) = 0$ hence $DP = 0$. For EO, we consider the conditional distributions of $Z|S = 0, Y = y$ and $Z|S = 1, Y = y$. Since $Z \perp\!\!\!\perp S$, it follows that $P(Z|S = s, Y = y) = P(Z|Y = y), s \in \{0, 1\}$. Therefore the distributions are identical across groups for each $y$, thus implying $EO = 0$. The same argument applies to $EOpp$. $\square$

Although achieving independence between $Z$ and $S$ ($Z \perp\!\!\!\perp S$) is challenging in practice, noising the embedding subspace encoding relevant information of $S$ provides a feasible way towards the target. Assuming that the data embedding $Z$ can be decomposed into two the task ($Z^t$) and sensitive ($Z^s$) embeddings, $Z = (Z^t, Z^s)$, where $Z^t \perp\!\!\!\perp S$. If $Z^s$ is obfuscated by a large amount of noise such that it is a pure random embedding $N$ with $N \perp\!\!\!\perp S$, then it would imply $Z \perp\!\!\!\perp S$. As $\delta_{TV}$ upper-bounds the membership inference accuracy (Theorem 3.1 Aubinais et al. (2023)), we also expect lower prediction accuracy on the attribute $S$ from $Z$. Our design of the FairNVT framework follows from such intuitions.

## C  EXPERIMENT SETUP DETAILS

**Architectures and Implementations.**  We use the ViT-Base model [6] as the frozen backbone for the CelebA and UTKFace datasets. Both task and sensitive adapters are bottleneck adapters inserted into each Transformer block of the frozen backbone. Each adapter consists of a down-projection that maps hidden states to a lower-dimensional space and an up-projection that restores them to the original hidden dimension. The reduction factor is a tunable hyperparameter. The task and sensitive classification heads are Multi-Layer Perceptrons (MLPs) whose hidden layer size matches the respective embedding dimension; the number of hidden layers is also treated as a tunable hyperparameter. For evaluating representation-level fairness via attacker accuracies, we use an attacker network with the same architecture as the task classification head. Table 5 summarizes an example FairNVT architecture used in the experiment for the task attribute *expression (smiling)* and the sensitive attribute *gender (male)*. FairNVT for vision task trains only 5.4M parameters ($\sim 6\%$ of ViT-Base) by freezing the backbone and introducing lightweight adapters and classification heads, significantly reducing computational cost compared to full fine-tuning.

Table 5: FairNVT architectural specifications for the experiment with task attribute *expression (smiling)* and sensitive attribute *gender (male)*. Layer dimensions are denoted as $N_{\text{weight\_in}} \times N_{\text{weight\_out}} + N_{\text{bias}}$. Task and sensitive adapter layers are attached after the final dense layer of the frozen ViT encoder (output dimension = 768) in all 11 encoder layers. Noise injection and embedding concatenation introduce no trainable parameters.

| Architecture | Layer | Specification | Output Size |
|---|---|---|---|
| Task Adapter | down_projection | $(768 \times 96 + 96) \times 11$ | 96 |
| | up_projection | $(96 \times 768 + 768) \times 11$ | 768 |
| Sensitive Adapter | down_projection | $(768 \times 48 + 48) \times 11$ | 48 |
| | up_projection | $(48 \times 768 + 768) \times 11$ | 768 |
| Noise Injection | \ | \ | 768 |
| Embedding Concatenation | \ | \ | $768 \times 2$ |
| Task Clf Head | linear_0 | $(768 \times 2) \times (768 \times 2) + (768 \times 2)$ | $768 \times 2$ |
| | tanh_activation | \ | $768 \times 2$ |
| | linear_1 | $(768 \times 2) \times 2 + 2$ | 2 |
| Sensitive Clf Head | linear_0 | $768 \times 768 + 768$ | 768 |
| | tanh_activation | \ | 768 |
| | linear_1 | $768 \times 2 + 2$ | 2 |

**Implementation Details.**  We implement all models in PyTorch Paszke et al. (2019) and train them on a workstation equipped with an AMD EPYC 7H12 CPU (64 cores) with a NVIDIA A100 GPU. We use the AdamW Loshchilov & Hutter (2019); Kingma & Ba (2015) optimizer with $\beta_1 = 0.9$, $\beta_2 = 0.999$, a weight decay of 0.01, and a batch size of 256. Training the debiasing framework for one run takes approximately 3 hours on a single A100 GPU. Inference requires 1.1 seconds for a batch of 256 samples.

During training, we perform grid-search hyperparameter tuning over the following ranges: adapter reduction factor $\{4, 8, 16\}$; number of hidden layers $\{0, 1, 2\}$; learning rates (searched by half orders of magnitude, e.g., $1e{-}1$, $5e{-}2$, $1e{-}2$, etc., until the best run is not at a boundary value); gradient-clipping thresholds $\{1, 10, 100\}$; noise levels $\{1, 5, 10\}$; and loss-weight coefficients $\beta \in \{0, 0.1, 0.3, 0.5, 1.0, 3.0\}$.

---

[6] `google/vit-base-patch16-224`

For evaluation, accuracy and balanced accuracy are computed from the predicted and true task labels. Fairness metrics (DP, EO, and EOpp) are computed using the predicted and true task labels together with the true sensitive attributes. The attacker setup follows Kumar et al. (2023): in an independent run, the attacker receives the task-classifier embeddings as input $X$ and the corresponding sensitive attributes $Y$ from the training and test sets. The attacker is trained to predict $Y$ from $X$ until the training accuracy no longer improves, and its test accuracy is reported as the attacker's ability to recover the sensitive attribute from the learned representation.

## D    MORE EXPERIMENT RESULTS

**More experiments on CelebA.**    Table 6 shows the results on more task, sensitive attribute pairs in the CelebA dataset. In most cases, we observe FairNVT showing a good balance between the prediction and fairness objectives, achieves better or comparable performances to the best baseline across different metrics.

**Additional qualitative results on CelebA.**    We provide additional samples for the *expression (smiling)* task with *gender (male)* as the sensitive attribute. Heatmaps are computed with *SmoothGrad* on the predicted class logit by averaging input gradients over 25 Gaussian noised, normalized inputs ($\sigma = 0.10$), aggregating $|\nabla_x|$ across channels, bilinearly resizing, and applying a light $3\times3$ blur, with maps normalized independently per image so intensities reflect within panel variation. Warmer regions indicate stronger contributions to the output logit. FairNVT concentrates on expression-relevant areas (e.g., mouth, cheeks), suggesting reduced reliance on gender-correlated cues and improved fairness via task-specific evidence.

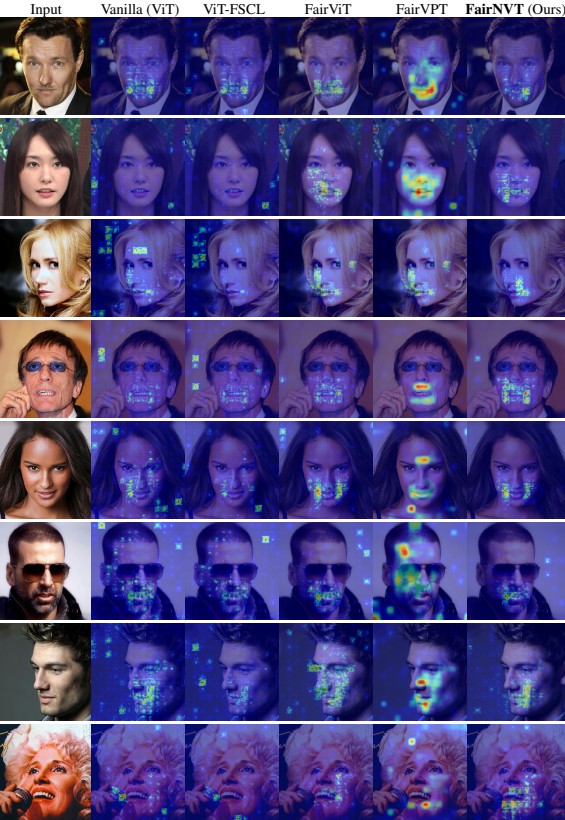

Figure 4: Additional examples: Gradient-based saliency map for the *expression (smiling)* as main task and *gender (male)* as sensitive attribute. Warmer regions indicate stronger contribution to the output logit. FairNVT primarily attends to expression-relevant areas (mouth/cheeks), demonstrating reduced reliance on gender-correlated cues.

Table 6: **Image-Based Classification task:** Comparing our method with baselines on CelebA Liu et al. (2015) dataset. All reported values are scaled by $\times 10^2$.

| Method | Acc($\uparrow$) | BAcc($\uparrow$) | DP($\downarrow$) | EOpp($\downarrow$) | EO($\downarrow$) | Att.Acc($\downarrow$) |
|---|---|---|---|---|---|---|
| Vanilla | $89.9_{\pm 0.1}$ | $89.4_{\pm 0.3}$ | $10.0_{\pm 0.6}$ | $2.1_{\pm 0.3}$ | $6.1_{\pm 1.2}$ | $87.8_{\pm 0.4}$ |
| ViT-FSCL | $88.7_{\pm 0.1}$ | $88.0_{\pm 0.1}$ | $7.4_{\pm 0.8}$ | $\mathbf{0.7_{\pm 0.2}}$ | $2.5_{\pm 0.1}$ | $87.4_{\pm 0.1}$ |
| FairViT | $\underline{92.5_{\pm 0.2}}$ | $91.9_{\pm 0.2}$ | $\mathbf{5.6_{\pm 0.3}}$ | $1.8_{\pm 0.9}$ | $\underline{2.3_{\pm 0.2}}$ | $\underline{86.2_{\pm 0.2}}$ |
| FairNVT(Ours) | $\mathbf{92.8_{\pm 0.1}}$ | $\mathbf{92.1_{\pm 0.1}}$ | $\underline{5.8_{\pm 0.3}}$ | $\underline{1.7_{\pm 1.1}}$ | $\mathbf{2.2_{\pm 0.2}}$ | $\mathbf{66.5_{\pm 0.1}}$ |

(a) Task: Expression; Sensitive Attribute: Age (Young)

| Method | Acc($\uparrow$) | BAcc($\uparrow$) | DP($\downarrow$) | EOpp($\downarrow$) | EO($\downarrow$) | Att.Acc($\downarrow$) |
|---|---|---|---|---|---|---|
| Vanilla | $\underline{81.6_{\pm 0.2}}$ | $63.2_{\pm 0.2}$ | $33.1_{\pm 2.7}$ | $40.3_{\pm 3.0}$ | $36.8_{\pm 5.2}$ | $98.8_{\pm 0.1}$ |
| ViT-FSCL | $80.4_{\pm 1.1}$ | $64.8_{\pm 0.0}$ | $24.7_{\pm 0.2}$ | $35.2_{\pm 0.1}$ | $24.7_{\pm 0.2}$ | $97.8_{\pm 0.1}$ |
| FairViT | $\mathbf{81.9_{\pm 0.3}}$ | $66.9_{\pm 0.4}$ | $\underline{20.4_{\pm 0.5}}$ | $30.6_{\pm 1.2}$ | $\underline{19.8_{\pm 0.9}}$ | $92.0_{\pm 0.4}$ |
| FairNVT(Ours) | $81.2_{\pm 0.1}$ | $\mathbf{67.4_{\pm 0.5}}$ | $\mathbf{8.1_{\pm 0.6}}$ | $\mathbf{8.2_{\pm 1.8}}$ | $\mathbf{8.3_{\pm 1.8}}$ | $\mathbf{55.8_{\pm 0.9}}$ |

(b) Task: Big Nose; Sensitive Attribute: Gender (Male)

| Method | Acc($\uparrow$) | BAcc($\uparrow$) | DP($\downarrow$) | EOpp($\downarrow$) | EO($\downarrow$) | Att.Acc($\downarrow$) |
|---|---|---|---|---|---|---|
| Vanilla | $84.4_{\pm 0.6}$ | $81.2_{\pm 0.5}$ | $10.3_{\pm 1.0}$ | $8.5_{\pm 1.0}$ | $7.7_{\pm 2.0}$ | $87.7_{\pm 0.4}$ |
| ViT-FSCL | $83.3_{\pm 0.6}$ | $77.9_{\pm 2.0}$ | $\mathbf{5.3_{\pm 0.8}}$ | $\mathbf{2.0_{\pm 0.4}}$ | $\mathbf{1.3_{\pm 0.3}}$ | $87.4_{\pm 0.1}$ |
| FairViT | $\underline{86.6_{\pm 0.4}}$ | $\underline{83.7_{\pm 0.3}}$ | $9.0_{\pm 0.5}$ | $3.5_{\pm 1.6}$ | $2.8_{\pm 0.6}$ | $\underline{86.4_{\pm 0.3}}$ |
| FairNVT(Ours) | $\mathbf{87.0_{\pm 0.5}}$ | $\mathbf{84.0_{\pm 0.5}}$ | $\underline{7.1_{\pm 0.7}}$ | $\underline{3.4_{\pm 0.9}}$ | $\underline{2.4_{\pm 0.5}}$ | $\mathbf{66.8_{\pm 0.1}}$ |

(c) Task: Wavy hair; Sensitive Attribute: Age(Young)

| Method | Acc($\uparrow$) | BAcc($\uparrow$) | DP($\downarrow$) | EOpp($\downarrow$) | EO($\downarrow$) | Att.Acc($\downarrow$) |
|---|---|---|---|---|---|---|
| Vanilla | $\underline{99.1_{\pm 0.1}}$ | $94.1_{\pm 0.3}$ | $\underline{10.8_{\pm 0.1}}$ | $5.3_{\pm 1.5}$ | $4.3_{\pm 1.3}$ | $98.7_{\pm 0.1}$ |
| ViT-FSCL | $\underline{99.1_{\pm 0.1}}$ | $95.0_{\pm 0.2}$ | $10.8_{\pm 0.4}$ | $3.9_{\pm 0.4}$ | $2.4_{\pm 0.2}$ | $97.9_{\pm 0.2}$ |
| FairViT | $99.0_{\pm 0.2}$ | $98.0_{\pm 0.2}$ | $\mathbf{10.0_{\pm 0.3}}$ | $\mathbf{0.8_{\pm 0.3}}$ | $\mathbf{0.6_{\pm 0.3}}$ | $\underline{97.6_{\pm 0.2}}$ |
| FairNVT(Ours) | $\mathbf{99.6_{\pm 0.0}}$ | $\mathbf{98.7_{\pm 0.0}}$ | $11.2_{\pm 0.1}$ | $\underline{0.7_{\pm 0.4}}$ | $\underline{0.7_{\pm 0.4}}$ | $\mathbf{53.8_{\pm 0.3}}$ |

(d) Task: Wearing glasses; Sensitive Attribute: Gender (Male)

| Method | Acc($\uparrow$) | BAcc($\uparrow$) | DP($\downarrow$) | EOpp($\downarrow$) | EO($\downarrow$) | Att.Acc($\downarrow$) |
|---|---|---|---|---|---|---|
| Vanilla | $\underline{99.1_{\pm 0.1}}$ | $95.4_{\pm 0.2}$ | $13.7_{\pm 0.2}$ | $7.0_{\pm 0.6}$ | $6.3_{\pm 1.6}$ | $88.1_{\pm 0.2}$ |
| ViT-FSCL | $99.0_{\pm 0.1}$ | $95.1_{\pm 1.1}$ | $13.1_{\pm 0.5}$ | $5.9_{\pm 0.1}$ | $3.3_{\pm 0.2}$ | $\underline{87.5_{\pm 0.1}}$ |
| FairViT | $\underline{99.1_{\pm 0.3}}$ | $\mathbf{97.4_{\pm 0.4}}$ | $\underline{13.0_{\pm 0.7}}$ | $\mathbf{2.7_{\pm 0.6}}$ | $\underline{2.9_{\pm 0.5}}$ | $89.4_{\pm 0.6}$ |
| FairNVT(Ours) | $\mathbf{99.3_{\pm 0.2}}$ | $96.9_{\pm 0.5}$ | $\mathbf{12.4_{\pm 1.0}}$ | $\underline{3.0_{\pm 1.1}}$ | $\mathbf{2.8_{\pm 1.3}}$ | $\mathbf{67.3_{\pm 0.1}}$ |

(e) Task: Wearing Glasses; Sensitive Attribute: Age (Young)

| Method | Acc($\uparrow$) | BAcc($\uparrow$) | DP($\downarrow$) | EOpp($\downarrow$) | EO($\downarrow$) | Att.Acc($\downarrow$) |
|---|---|---|---|---|---|---|
| Vanilla | $85.5_{\pm 0.3}$ | $85.2_{\pm 0.4}$ | $9.6_{\pm 0.4}$ | $4.7_{\pm 0.6}$ | $4.6_{\pm 0.9}$ | $98.7_{\pm 0.0}$ |
| ViT-FSCL | $82.6_{\pm 0.5}$ | $81.8_{\pm 0.6}$ | $\underline{7.5_{\pm 2.0}}$ | $\underline{1.3_{\pm 0.7}}$ | $2.5_{\pm 1.7}$ | $97.6_{\pm 0.1}$ |
| FairViT | $\underline{93.4_{\pm 0.1}}$ | $93.3_{\pm 0.2}$ | $9.0_{\pm 0.4}$ | $1.6_{\pm 0.3}$ | $\underline{1.5_{\pm 0.4}}$ | $96.1_{\pm 0.4}$ |
| FairNVT(Ours) | $\mathbf{93.7_{\pm 0.1}}$ | $\mathbf{93.7_{\pm 0.1}}$ | $\mathbf{6.3_{\pm 0.8}}$ | $\mathbf{0.9_{\pm 0.1}}$ | $\mathbf{1.5_{\pm 0.6}}$ | $\mathbf{52.4_{\pm 0.6}}$ |

(f) Task: Mouth Slightly Open; Sensitive Attribute: Gender(Male)

| Method | Acc($\uparrow$) | BAcc($\uparrow$) | DP($\downarrow$) | EOpp($\downarrow$) | EO($\downarrow$) | Att.Acc($\downarrow$) |
|---|---|---|---|---|---|---|
| Vanilla | $84.7_{\pm 1.8}$ | $83.9_{\pm 1.7}$ | $7.4_{\pm 1.5}$ | $1.8_{\pm 1.3}$ | $6.1_{\pm 1.8}$ | $85.2_{\pm 3.9}$ |
| ViT-FSCL | $83.8_{\pm 0.1}$ | $82.9_{\pm 0.1}$ | $\underline{5.7_{\pm 1.2}}$ | $1.5_{\pm 1.3}$ | $2.3_{\pm 1.1}$ | $87.4_{\pm 0.1}$ |
| FairViT | $\underline{93.4_{\pm 0.3}}$ | $93.1_{\pm 0.2}$ | $7.0_{\pm 0.3}$ | $\underline{0.8_{\pm 0.2}}$ | $\underline{0.9_{\pm 0.3}}$ | $\underline{82.3_{\pm 0.2}}$ |
| FairNVT(Ours) | $\mathbf{94.0_{\pm 0.0}}$ | $\mathbf{93.7_{\pm 0.2}}$ | $\mathbf{4.6_{\pm 0.2}}$ | $\mathbf{0.3_{\pm 0.1}}$ | $\mathbf{0.6_{\pm 0.1}}$ | $\mathbf{65.8_{\pm 0.1}}$ |

(g) Task: Mouth Slightly Open; Sensitive Attribute: Age(Young)

**Qualitative results on BIOS.** We evaluate fairness by comparing predictions on pairs of sentences that are identical except for words that indicate gender. Table 13 summarizes how the predicted profession probabilities change under these minimal substitutions. The vanilla model shows substantial shifts, whereas FairNVT produces more stable predictions across sentences that differ only in gender-indicative terms.

# E    MORE ABLATION RESULTS

**Representation-level fairness results with stronger attacker models.**    Table 7 summarizes results obtained with stronger attacker models. Increasing the number of hidden layers in the MLP attacker does not substantially affect its accuracy (Att. Acc.) in predicting the sensitive attribute from the de-biased embeddings, indicating that clipping and noise injection effectively mitigate sensitive-attribute leakage. When the sensitive attribute is imbalanced and binary, the raw attacker accuracy can be misleading since predicting the majority class yields higher scores. To address this, we also report balanced attacker accuracy (Balanced Att. Acc.). In experiments where *age (young)* serves as an imbalanced sensitive attribute, the balanced attacker accuracies confirm that FairNVT consistently reduces the attacker's success rate to near-random levels.

Table 7: **Performance with different attackers.** We evaluate the representation-level fairness result of FairNVT with stronger attacker models. All reported values are scaled by $\times 10^2$.

| Attacker Model | Task | # Hidden Layers | Att. Acc($\downarrow$) | Balanced Att. Acc($\downarrow$) |
|---|---|---|---|---|
| **MLP** | **Task: Expression (Smiling)** **Sens.: Gender (Male)** | 1 | 52.3 | 50.9 |
| | | 3 | 52.2 | 50.8 |
| | | 10 | 52.4 | 51.1 |
| | **Task: Big Nose** **Send.: Age (Young)** | 1 | 68.1 | 53.2 |
| | | 3 | 67.9 | 53.3 |
| | | 10 | 68.3 | 53.6 |

**Comparing different task classifier inputs.**    Table 8 shows the effect on accuracy and fairness metrics when the task classifier input changes. These results are trained with the same loss $L$ as in Section 3, except nullifying the orthogonality loss when the sensitive embedding ($e^s$) is not present. When using the backbone frozen embedding $h$ directly (row 1) or concatenating task ($e_t$) and sensitive embedding ($e_s$) without noise (row 2), it is more difficult to obtain fair outcomes when the sensitive information is not obfuscated, indicated by higher DP, EOpp, EO and Att.Acc values. Naively adding noise to $h$ (row 3) could achieve good fairness outcomes but hurting the task performance. Although concatenating pure noise $z$ with the task embedding $e_t$ (row 4) slightly improves fairness metrics, it does not achieve the same level of fairness as FairNVT. Since $z$ is not injected through the sensitive branch, it does not target and suppress sensitive information directly. We show in Table 7(App. E) that increasing attacker model complexity by adding more layers does not improve the attacker accuracy in FairNVT. We additionally test on alternative ways of fusing $e_s, e_t$. Aligning noise $z$ with $e_s$ before concatenating with $e_t$ (row 5) improves task accuracy and achieves competitive DP, EOpp and EO, but leaks sensitive information as attacker accuracy increases. Fusing task and sensitive embedding with self-attention (row 6) preserves more sensitive information thus slightly hurting the fairness outcomes. Overall, simple concatenation of noisy $e_s$ with $e_t$ achieves the balance between accurate task prediction, fair outcomes and reducing sensitive information leakage.

Table 8: **Effect of noise and projection choices on fairness and utility.** We assess variants of task classification head inputs constructed from frozen backbone output without having any adapter $h$, sensitive embedding $e_s$, task embedding $e_t$, and injected noise $z$. The comparison highlights how noise injection, projection, and attention choices influence task performance and fairness. All reported values are scaled by $\times 10^2$. Task: Expression (Smiling); Sensitive attribute: Gender (Male).

| $Task^{clf}$ Inputs | Acc($\uparrow$) | BAcc($\uparrow$) | DP($\downarrow$) | EOpp($\downarrow$) | EO($\downarrow$) | Att.Acc($\downarrow$) |
|---|---|---|---|---|---|---|
| $[h]$ | 90.2$\pm$0.0 | 90.1$\pm$0.1 | 10.5$\pm$1.3 | 1.5$\pm$0.7 | 2.3$\pm$0.5 | 98.8$\pm$0.0 |
| $[e_s, e_t]$ | 92.9$\pm$0.3 | 92.9$\pm$0.2 | 10.1$\pm$0.4 | 2.4$\pm$0.4 | 3.0$\pm$0.4 | 98.5$\pm$0.1 |
| $[z, h]$ | 86.4$\pm$0.2 | 86.5$\pm$0.2 | 7.0$\pm$0.8 | 1.4$\pm$0.4 | 4.1$\pm$0.9 | 89.5$\pm$0.1 |
| $[z, e_t]$ | 93.0$\pm$0.1 | 93.0$\pm$0.1 | 10.0$\pm$0.5 | **0.5$\pm$0.3** | 2.8$\pm$0.5 | 63.7$\pm$0.6 |
| $[\frac{\langle z^i, e_s^i \rangle}{\|e_s^i\|} e_s^i, e_t]$ | 92.2$\pm$1.0 | **93.2 $\pm$0.9** | **9.8$\pm$0.8** | 0.8$\pm$0.5 | 2.1$\pm$0.5 | 98.8$\pm$0.0 |
| Attn $(e_s + z, e_t)$ | 91.9$\pm$0.2 | 91.7$\pm$0.2 | 10.2$\pm$1.1 | 1.3$\pm$0.5 | 2.8$\pm$1.0 | 54.5$\pm$0.2 |
| FairNVT | **93.1$\pm$0.2** | 93.0$\pm$0.3 | 9.9$\pm$0.3 | 0.8$\pm$0.3 | **1.5$\pm$0.5** | **51.6$\pm$0.4** |

**Effect of ablating model components.**    Table 9 presents ablation results for different model components. The results are consistent with the main findings in Section 4: the DP loss primarily drives fairness improvements, noise injection reduces sensitive-attribute leakage, and the orthogonality

loss further enhances fairness with minimal impact on task performance. Model components also exhibit interacting effects; in particular, combining DP loss with noise injection further decreases DP, EOpp, and EO scores, indicating that enhancing representation-level fairness can align with improvements in prediction-level fairness.

Table 9: **Ablation of FairNVT components on CelebA.** We toggle Demographic Parity loss (DP), Orthogonality loss (Orth), and Noise injection (Noise). ✓and ✗means the component is present and absent respectively. All reported values are scaled by $\times 10^2$.

| Task/Sens. | DP Loss | Orth Loss | Noise | Acc(↑) | BAcc (↑) | DP (↓) | EOpp(↓) | EO(↓) | Att Acc(↓) |
|---|---|---|---|---|---|---|---|---|---|
| **Task: Expression (Smiling)** | ✓ | ✗ | ✗ | 92.7 | 92.7 | 8.6 | 0.8 | 4.4 | 98.9 |
|  | ✗ | ✓ | ✗ | 93.4 | 93.1 | 14.3 | 4.0 | 4.0 | 99.0 |
| **Sens.: Gender (Male)** | ✗ | ✗ | ✓ | 93.0 | 92.7 | 14.8 | 4.8 | 4.8 | 54.2 |
|  | ✓ | ✓ | ✓ | 93.0 | 93.0 | 9.8 | 0.3 | 2.8 | 52.6 |
|  | ✗ | ✗ | ✗ | 83.2 | 69.7 | 23.7 | 20.7 | 20.7 | 87.8 |
|  | ✓ | ✗ | ✗ | 82.4 | 67.9 | 13.1 | 4.6 | 4.6 | 88.0 |
| **Task: Big Nose** | ✗ | ✓ | ✗ | 83.5 | 70.5 | 23.8 | 21.0 | 21.0 | 88.0 |
|  | ✗ | ✗ | ✓ | 83.0 | 69.8 | 23.4 | 18.9 | 18.9 | 70.6 |
| **Sens. Age (Young)** | ✗ | ✓ | ✓ | 83.2 | 71.0 | 24.9 | 19.8 | 19.8 | 69.8 |
|  | ✓ | ✗ | ✓ | 82.4 | 69.1 | 13.3 | 3.3 | 3.3 | 68.6 |
|  | ✓ | ✓ | ✗ | 82.6 | 68.4 | 12.7 | 3.8 | 3.8 | 88.0 |
|  | ✓ | ✓ | ✓ | 82.2 | 68.3 | 12.6 | 2.3 | 2.6 | 68.5 |

**Sensitivity of loss weight coefficients.** We analyze the effect of loss weight coefficients in Table 10, using the task and sensitive attributes *expression (smiling)* and *gender (male)*, respectively. A moderate orthogonality loss weight consistently achieves the best balance between task accuracy and fairness metrics, indicating that this setting effectively disentangles task and sensitive embeddings without degrading representation quality. Increasing the DP loss weight improves prediction-level fairness, particularly for demographic parity difference, which it directly optimizes, though with a gradual trade-off in task performance. Because EO and EOpp condition on specific label groups, they are naturally more sensitive to small prediction variations, yet we observe stable improvements at moderate DP weights. Overall, these trends highlight that the loss weights control the fairness–utility balance in a predictable manner, and tuning them allows FairNVT to adapt robustly across datasets and attribute combinations.

Table 10: **Sensitivity of loss weight coefficients.** We evaluate the performance of FairNVT when the loss weight coefficients changes. All reported values are scaled by $\times 10^2$. Task: Expression (Smiling); Sensitive attribute: Gender (Male).

|  | Level | Acc(↑) | BAcc(↑) | DP(↓) | EOpp(↓) | EO(↓) | Att.Acc(↓) |
|---|---|---|---|---|---|---|---|
| **Orth Loss** | 0 | 92.8 | 92.7 | 9.9 | 0.2 | 3.1 | 53.0 |
|  | 0.01 | 92.8 | 92.8 | 10.2 | 0.4 | 2.5 | 53.2 |
|  | 0.1 | 93.0 | 93.0 | 9.8 | 0.3 | 2.8 | 52.6 |
|  | 1.0 | 92.8 | 92.7 | 10.4 | 0.7 | 2.4 | 52.1 |
| **DP Loss** | 0 | 93.2 | 92.8 | 14.5 | 4.8 | 4.8 | 52.9 |
|  | 0.01 | 93.0 | 92.7 | 14.3 | 4.2 | 4.2 | 53.1 |
|  | 0.3 | 93.0 | 93.0 | 9.8 | 0.3 | 2.8 | 52.6 |
|  | 1.0 | 92.1 | 92.3 | 5.7 | 3.5 | 6.1 | 53.4 |

**Sensitivity of embedding clipping.** As discussed in Section 4, we clip the embeddings to an upper bound $C$ before adding noise, which helps control the obfuscation of sensitive information. We analyze the sensitivity of the model to different values of the clipping threshold $C$. Changing $C$ under a fixed noise multiplier ($\sigma$) has a combined effect: it alters the embedding magnitude while also changing the effective noise level, since the noise variance $\sigma^2 C^2$ scales with $C$ (Table 11, rows 1-3). In this setting, smaller $C$ values degrade representation-level fairness, as reflected by higher attacker accuracies. When controlling for noise variance (Table 11, rows 4-6), we observe that varying $C$ produces no significant change in either task accuracy or fairness metrics, suggesting that the clipping operation itself has limited influence once the noise scale is fixed.

Table 11: **Sensitivity of embedding clipping threshold.** We evaluate the performance of FairNVT when the clipping threshold changes. All reported values are scaled by $\times 10^2$. Task: Expression (Smiling); Sensitive attribute: Gender (Male).

| | Level | Acc(↑) | BAcc(↑) | DP(↓) | EOpp(↓) | EO(↓) | Att.Acc(↓) |
|---|---|---|---|---|---|---|---|
| **Clip Threshold** (with same noise multiplier) | 1 | 93.1 | 93.1 | 9.7 | 0.4 | 2.9 | 89.0 |
| | 10 | 93.0 | 93.0 | 9.8 | 0.3 | 2.8 | 52.6 |
| | 100 | 92.3 | 92.2 | 10.6 | 0.9 | 1.9 | 53.1 |
| **Clip Threshold** (with same noise amount) | 1 | 92.8 | 92.8 | 9.8 | 0.2 | 2.9 | 52.6 |
| | 10 | 93.0 | 93.0 | 9.8 | 0.3 | 2.8 | 52.6 |
| | 100 | 92.8 | 92.8 | 9.7 | 0.3 | 2.7 | 52.2 |

**Sensitivity of number of noise draw at inference.** Table 12 reports results obtained when varying the number of noise draws during inference. The task prediction accuracies from a single noise draw are nearly identical to those from multiple draws, indicating that sensitive information is effectively disentangled from task-relevant features and that perturbing the sensitive subspace does not substantially alter task predictions. While majority voting over multiple noisy embeddings slightly improves task accuracy, it also marginally increases DP, EOpp, EO, and attacker accuracies, suggesting that aggregating multiple de-biased embeddings reintroduces a small amount of sensitive information. Overall, when fairness certification (Appendix B) is not required, a single noise draw is sufficient to achieve strong task performance and fairness outcomes.

Table 12: **Sensitivity of number of noise draw at inference.** We evaluate the performance of FairNVT when the key hyperparameter value changes. All reported values are scaled by $\times 10^2$. Task: Expression (Smiling); Sensitive attribute: Gender (Male).

| | Level | Acc(↑) | BAcc(↑) | DP(↓) | EOpp(↓) | EO(↓) | Att.Acc(↓) |
|---|---|---|---|---|---|---|---|
| **Num. Noise Draw** (Inference Time) | 1 | 93.0 | 93.0 | 9.8 | 0.3 | 2.8 | 52.6 |
| | 10 | 93.2 | 93.2 | 9.9 | 0.4 | 3.3 | 53.3 |
| | 50 | 93.5 | 93.4 | 10.3 | 0.5 | 2.8 | 53.1 |

Table 13: **Qualitative BIOS examples.** We show pairs of biography snippets that differ only in gender indicators. For each snippet, we display the model's predicted occupation and the prediction score for the ground-truth label. Vanilla model predictions vary substantially across genders, suggesting reliance on gender cues, whereas FairNVT yields more stable scores and consistent predictions, indicating improved robustness to gender indicators.

| ID | BIO Snippet | Vanilla | FairNVT |
|---|---|---|---|
| 1 | He specializes in development economics, household economics, and personnel economics. In 2003 he received his Ph.D. in Economics from the London School of Economics... | professor (0.903) | professor (0.992) |
| 1 | She specializes in development economics, household economics, and personnel economics. In 2003 she received her Ph.D. in Economics from the London School of Economics... | professor (0.882) | professor (0.993) |
| 2 | Prosper was born and raised in Miami Beach, FL. He received his Bachelor's degree from Emory University and graduated with honors from the University of Miami School of Law... | attorney (0.971) | attorney (0.971) |
| 2 | Prosper was born and raised in Miami Beach, FL. She received her Bachelor's degree from Emory University and graduated with honors from the University of Miami School of Law... | attorney (0.939) | attorney (0.970) |
| 3 | She has been travelling the world, and worked, amongst others, on a documentary photography project in India with an orphanage... | photographer (0.643) | photographer (0.908) |
| 3 | He has been travelling the world, and worked, amongst others, on a documentary photography project in India with an orphanage... | photographer (0.729) | photographer (0.864) |
| 4 | She studied at EFET Paris and NYU New-York respectively. While working in a post-production, she develops her own photographic concept... | photographer (0.664) | photographer (0.966) |
| 4 | He studied at EFET Paris and NYU New-York respectively. While working in a post-production, he develops his own photographic concept... | photographer (0.804) | photographer (0.941) |
| 5 | He attended the University of California, San Francisco (UCSF), School of Medicine and subsequently trained at Children's Hospital Los Angeles for residency... | physician (0.717) | physician (0.818) |
| 5 | She attended the University of California, San Francisco (UCSF), School of Medicine and subsequently trained at Children's Hospital Los Angeles for residency... | physician (0.826) | physician (0.755) |

