# OpenReview forum: "Improving Fairness via Noise Injection in Vision Transformers"
_ICLR.cc/2026/Workshop/AFAA — AFAA 2026 Poster_

### Official Review · Reviewer_mGch · 2026-02-19
**Simultaneously addressing prediction- and representation-level fairness: efficiently debiasing pretrained transformers encoders despite the potential cost of hyperparameter search.**

**Rating:** 5
**Confidence:** 2

**Summary:**

The paper proposes to address fairness simultaneously at two levels: prediction where the model's goal is to have outputs independent of the sensitive attribute; and representation level where it is crucial for learned embeddings to not encode sensitive information. This is achieved by adding light-weight (in terms of computational costs) adapters to extract task-relevant and sensitive information from the potentially biased embedding. Then  random calibrated Gaussian noise is applied to the sensitive subspace, which is in turn fused with the task representation; together with orthogonality constraints and demographic-parity regularization. The approach's success in reducing sensitive-attribute leakage in the learned embeddings and encouraging fairer downstream predictions is validated in the experimental section via three imaging- and text-based classification tasks, and across eight baselines. In addition to this, related works and mathematical details are discussed in appendix.

**Strengths:**

1-) Various evaluation metrics and compatibility with a good number of pretrained transformer encoders.
2-) Ablation studies discuss  the contribution of different components in the proposed method (Adapters and classification heads, Noise injection and embedding fusion) as well as the influence of noise strength. Even more ablation studies are presented in the appendix.
3-) The connection between randomized calibrated Gaussian smoothing and  fairness is novel to the best of my knowledge. Moreover, the authors addresses complementary issues that are otherwise usually individually treated.
4-) The mathematical details of the framework are provided in the appendix
5-) Efficient training and interpretable algorithm.

**Weaknesses:**

1-) Orthogonality loss encourages separation, but to the best of my knowledge, it may not always be obtainable.
2-) Even though the adapters themselves are light-weight, the procedure uses grid search over wide ranges hyperparameters. This may end up being expensive.

---

### Official Review · Reviewer_msQL · 2026-02-20
**Noise-Calibrated Subspace Debiasing for Fair Representation Learning**

**Rating:** 4
**Confidence:** 3

**Summary:**

This paper proposes FairNVT, a lightweight representation-level debiasing framework designed for pretrained transformer models. The core idea is to learn a sensitive-attribute subspace via an adversarial objective and then suppress this information by injecting calibrated Gaussian noise during training. The approach is inspired by randomized smoothing and aims to reduce sensitive information leakage while preserving downstream task performance.

The method is implemented as a small adapter module, allowing the backbone model to remain frozen. Experiments are conducted across vision and text tasks, comparing FairNVT against several representation-level and adversarial fairness baselines. Results show improved fairness–utility trade-offs, reduced attribute leakage (measured via attacker accuracy), and robustness across varying noise levels.

**Strengths:**

- The lightweight implementation and its compatibility with pretrained transformers might enhance its practical applicability.


- The paper evaluates across modalities (vision and text) and reports both fairness and utility metrics, including attacker-based leakage analysis. The ablation studies on noise scale strengthen the empirical analysis.

**Weaknesses:**

- The approach does not establish formal fairness guarantees nor derive theoretical bounds quantifying the extent of sensitive attribute removal.

- Experiments focus on moderate-scale models and primarily binary sensitive attributes. It is unclear how the approach scales to large foundation models or multi-attribute settings.

---

### Official Review · Reviewer_6sS7 · 2026-02-21
**Representation and Prediction Fairness via Adapters and Noise Injection: effective solution with potential inference cost**

**Rating:** 4
**Confidence:** 4

**Summary:**

The paper proposes a debiasing training framework built on top of pretrained transformers to achieve both representation-level and prediction-level fairness. The proposed method, FairVNT, consists of two adapters (one for sensitive attribute representations and one for task representations), a noise injection module, and a fusion embedding with two prediction heads (one for sensitive attributes and one for the main classification task).

The frozen pretrained representations are passed through the two adapters separately. The sensitive attribute representations are perturbed with Gaussian noise before being combined with the task adapter representations through a fusion layer for class prediction. The sensitive attribute prediction head operates directly on the sensitive adapter representations. Training minimizes a combination of losses, including the task classification loss, an orthogonality loss to reduce correlation between sensitive and task representations, and a demographic parity loss to encourage fair predictions.

The experimental evaluation considers both image classification using pretrained ViT and text classification using pretrained BERT. The results show that FairVNT simultaneously improves balanced accuracy, multiple fairness metrics (Equalized Odds, Demographic Parity, Equal Opportunity), and reduces sensitive information leakage compared to vanilla models and several debiasing baselines.

**Strengths:**

- The paper is well structured and generally easy to follow, despite the number of components
- FairVNT is effective at improving fairness metrics while also learning representations that contain less sensitive attribute information.
- The combination of architectural separation (dual adapters), noise injection, and losses is an interesting combination of previous literature to solve the fair representation problem and appears to work across modalities.

**Weaknesses:**

1. **Inference cost and efficiency.**
   The framework introduces two adapters and relies on multiple noise samples to produce fused embeddings, which increases computational cost and latency at inference time. A discussion or empirical evaluation of efficiency trade-offs would strengthen the paper.


2. **Ablation completeness.**
   It is unclear why the sensitive attribute prediction loss is not included in the ablation studies.


3. **Task dependence of debiased representations.**
   The learned debiased representations appear to remain task-specific. It would be helpful to clarify this point, as the debiasing objective is applied during training on the task dataset rather than aiming to produce representations that are universally debiased across multiple downstream tasks. The latter appears to be a different problem.

---

### Meta-Review · Area_Chair_izgX · 2026-02-26

**Recommendation:** Main Papers Track
**Confidence:** 5

**Metareview:**

The paper proposes FairVNT, a lightweight debiasing framework for pretrained transformers. Reviewers find the paper well written and empirically strong across both vision and text tasks, reporting improved fairness–utility trade-offs and reduced sensitive-attribute leakage relative to multiple baselines. The randomized-smoothing-inspired noise injection and the combined architectural/optimization design are viewed as notable contributions. Reviewers’ remaining concerns are mainly practical and scope-related, but they do not undermine the main claims. All reviews are positive, and I recommend acceptance.

---

### Decision · Program_Chairs · 2026-03-02

Accept (Poster)